# Neural Circuitry Polarization in the Spinal Dorsal Horn (SDH): A Novel Form of Dysregulated Circuitry Plasticity during Pain Pathogenesis

**DOI:** 10.3390/cells13050398

**Published:** 2024-02-25

**Authors:** Xufeng Chen, Shao-Jun Tang

**Affiliations:** Stony Brook University Pain and Anesthesia Research Center (SPARC), Department of Anesthesiology, Stony Brook University, Stony Brook, NY 11794, USA; xufeng.chen@stonybrookmedicine.edu

**Keywords:** chronic pain, gp120, NCP, circuitry plasticity

## Abstract

Pathological pain emerges from nociceptive system dysfunction, resulting in heightened pain circuit activity. Various forms of circuitry plasticity, such as central sensitization, synaptic plasticity, homeostatic plasticity, and excitation/inhibition balance, contribute to the malfunction of neural circuits during pain pathogenesis. Recently, a new form of plasticity in the spinal dorsal horn (SDH), named neural circuit polarization (NCP), was discovered in pain models induced by HIV-1 gp120 and chronic morphine administration. NCP manifests as an increase in excitatory postsynaptic currents (EPSCs) in excitatory neurons and a decrease in EPSCs in inhibitory neurons, presumably facilitating hyperactivation of pain circuits. The expression of NCP is associated with astrogliosis. Ablation of reactive astrocytes or suppression of astrogliosis blocks NCP and, concomitantly, the development of gp120- or morphine-induced pain. In this review, we aim to compare and integrate NCP with other forms of plasticity in pain circuits to improve the understanding of the pathogenic contribution of NCP and its cooperation with other forms of circuitry plasticity during the development of pathological pain.

## 1. Introduction to the Nociceptive Pathway

Nociception is a protective process that helps to prevent damage by noxious environmental stimuli, such as intense thermal and mechanical stimuli, as well as endogenous chemical irritants. Nociceptive stimuli are detected and transduced into electrical signals by spatialized periphery sensory neurons (nociceptors), which are situated in the dorsal root ganglia (DRG) and trigeminal ganglia [1,2]. Primary afferent nociceptors project to the spinal cord dorsal horn (SDH), where nociceptive signals are processed by projection neurons and interneurons. Projection neurons convey nociceptive information to the thalamus, the brain stem, and the cortex (Figure 1). 

The SDH and trigeminal nucleus are the first CNS relay stations of nociceptive information. The SDH is organized into six distinct laminae [3]. Nociceptive signals are conveyed to neurons in lamina I and outer lamina II via fine myelinated (Aδ) and unmyelinated (C) fibers, whereas innocuous signals are transmitted to neurons in laminae III via low-threshold Aβ fibers (Figure 2). The major targets of primary afferent neurons are projection neurons and interneurons. While projection neurons synapse at supraspinal and cortical regions, interneurons make synaptic connections locally with other neurons in the SDH, including projection neurons. Interneurons are classified into islet, central, radial and vertical cells based on their morphology. Most islet cells are GABAergic, whereas radial and vertical cells are predominantly glutamatergic. Central cells include both excitatory and inhibitory subgroups [4]. Patch clamp recordings show that C fibers innervate all four types of interneurons, whereas Aδ afferents only project to vertical and radial cells. Projection neurons are located mainly at laminae I and deeper laminae III-V. Around 80% of the projection neurons in laminae I express the neurokinin 1 (NK1) receptor [5]. These neurons are densely innervated by C and Aδ fibers, which release substance P following nociceptive stimuli [5,6,7,8] (Figure 2).

## 2. Principal Synaptic Receptors in Pain Neural Circuits

Glutamate is the predominant excitatory neurotransmitter in the SDH. It is sensed by two classes of receptors: ionotropic glutamate receptors, such as NMDA and AMPA receptors, and metabotropic glutamate receptors, such as mGluR1 and mGluR5 [9]. These receptors have been shown to play key roles in the pathogenesis of chronic pain [10,11,12] (Figure 3).

NMDA receptors are Mg^2+^-gated Ca^2+^ channels that are usually made up of two GluN1 subunits and two GluN2/GluN3 subunits. NMDA subunits are differentially regulated in different pain models. For example, formalin upregulated GluN2A and downregulated GluN2C, whereas nerve injury increases the expression of GluN2B but decreases that of GluN1 [13].

AMPA receptors are Na^+^ channels that consist of four subunits: GluA1–4. When the channel is lacking GluA2, it becomes Ca^2+^-permeable. GluA1 and GluA2 are expressed in the superficial lamina of the SDH. However, their expression patterns are different. GluA2-containing AMPA receptors are essentially present at all glutamatergic synapses in this region, whereas the GluA1 subunit is not contained in NK1 projection neurons [14,15,16].

Spinal cord injury causes upregulation of AMPA receptor expression in the superficial laminae of the spinal cord and mechanical and thermal hyperalgesia [17]. GluA1 and GluA2 subunits seem to contribute to the development of chronic pain. Mice lacking GluA1 but not GluA2 in DRG neurons show reduced mechanical hypersensitivity in models of chronic inflammatory pain [18], whereas GluA2-deficient mice showed enhanced long-lasting inflammatory hyperalgesia [19]. An AMPA receptor subunit switch is implicated in the development of chronic pain [20]. A substantial switch from GluA2- to GluA1-containing AMPAR was observed in the spinal cord after induction of peripheral inflammation [21,22,23]. The switch may promote the formation of Ca^2+^-permeable AMPARs and thus facilitate the activation of Ca^2+^-dependent pathways critical for central sensitization [9,21,24,25].

Metabotropic glutamate receptors are classified into three groups. Group I mGlu receptors (mGluR1 and 5) are predominantly localized postsynaptically. Group II (mGluR2 and 3) are both pre- and postsynaptic. Group III (mGluR 4, 6, 7 and 8) are presynaptic [26]. Most metabotropic glutamate receptors, except mGluR6 and mGluR8, are expressed in the spinal cord [27,28,29,30,31,32,33] (Figure 3). Spinal cord injury upregulates mGlu1 receptor expression, specifically on projection neurons of the spinothalamic tracts [34] and downregulates group II mGluR [35]. Non-selective mGluR agonist (1S, 3R)-ACPD facilitates neuronal responses to noxious stimulation in normal animals but inhibits those in inflamed animals [36]. In addition, intrathecal administration of the mGluR1 antagonist or mGluR1 knockdown in spinal cord reduces hyperalgesia and allodynia [37,38,39]. 

Inhibitory interneurons predominantly use GABA and/or glycine as their neurotransmitter. Many interneurons co-express these inhibitory neurotransmitters (Figure 3). 

There are two subtypes of GABA receptors: GABAA (ionotropic) and GABAB (metabotropic) receptors. Both are found in the terminals of nociceptors as well as postsynaptic receptors in spinal neurons [26,40,41]. Downregulation of the GABA-synthesizing enzyme [42], pre- and postsynaptic GABA receptors [43], or dysregulation of Cl^−^ homeostasis [44,45] in the spinal dorsal horn were reported to critically contribute to the development of neuropathic pain. 

Glycine receptors are pentameric complex. They are composed of two types of subunits: α(1–4) and β [46]. GlyR α3 deficient mice show reduced thermal and mechanical hyperalgesia, especially during the later phases of inflammation [47]. 

## 3. Glial Cells in Pain Neural Circuits

Glial cells include oligodendrocytes, astrocytes, and microglia. Their contribution to the pathogenesis of pain induction and maintenance has been identified [48,49,50,51,52].

Microglia are macrophage-like cells in the CNS. In response to nerve injury, microglia in the spinal cord undergo microgliosis, which manifests as a complex set of biological changes of microglia, including a morphological switch from ramified to amoeboid [53,54,55], rapid proliferation [54,56,57,58] and the release of a series of signaling molecules [55,59,60,61]. It has been shown that microgliosis regulated by FKN/CX3XR1 signaling is implicated in synaptic degeneration observed in patients with HIV-associated pain [61].

In addition to engulfing synapses, microglia also modulate synaptic transmission via releasing cytokine, chemokine and trophic factors (Figure 3). Microglial TNFα has been shown to enhance synaptic efficacy by increasing glutamate release in lamina II interneurons. Single-cell PCR analysis indicates that this effect is exclusively in excitatory interneurons that express vesicular glutamate transporter-2 (vGluT2) [62]. Meanwhile, TNFα recruited Ca^2+^-permeable AMPA receptors to dorsal horn neurons, resulting in increased sEPSC amplitude [63]. Similar to TNFα, glial IL-1β also enhances excitatory synaptic transmission, indicated by increased sEPSC frequency and amplitude [64]. In addition, TNFα and IL-1β increase NMDA receptor activity through stimulating phosphorylation of the NR1 subunit [65,66].

Following nerve injury, ATP released by peripheral afferents activates P2X4 receptors in microglia [67]. This process induces the release of BDNF, which is sufficient to induce pain behavior [67,68,69]. BDNF results in pain hypersensitivity by interrupting the chloride homeostasis of lamina I projection neurons. Specifically, BDNF decreases the expression of the potassium-chloride cotransporter 2 (KCC2), leading to a rise in intracellular chloride in these neurons [70,71] (Figure 3). This shift causes a switch of GABAA receptors from inhibition to excitation [70,72,73], resulting in the disinhibition of nociceptive processing. In the same model, however, it was reported that intrathecal BDNF shows a short-lasting anti-nociceptive effect, presumably due to the BDNF activity in rapid and reversible facilitation of GABA release in the SDH [74,75,76]. 

Astrocytes also play an important role in the pathogenesis of neuropathic pain [77]. Robust astrogliosis (indicated by increased GFAP expression) has been observed in various animal pain models, including neuropathic pain, spinal cord injury pain, inflammatory pain and HIV-associated pain [77,78,79,80,81,82]. Ablation of reactive astrocytes, inhibition of the spinal astrogliosis, or spinal knockdown of GFAP expression reduces pain development in animal models [51,52,83,84]. Astrogliosis in the SDH of HIV patients is specifically associated with the development of neuropathic pain, suggesting a pathogenic contribution of astrocytes to pain development in humans [85].

Astrocytes have extensive contacts with synapses, enabling them to regulate the local interstitial ionic and chemical environment during synaptic transmission [86,87]. Inhibition of glutamine uptake is sufficient to prevent the development of central sensitization [88]. Expression of glutamate transporters GLT-1 and GlAST is observed in the SDH neurons of neuropathic pain models [89], and spinal administration of riluzole, a positive regulator of glutamate transporter activity, attenuates neuropathic pain [90]. 

Astrocytes in the spinal cord release gliotransmitters such as D-serine and ATP. D-serine facilitates nociceptive responses of spinal wide dynamic range (WDR)neurons by activating NMDA receptors [91]. ATP affects synaptic transmission through the ionotropic P2X and metabotropic P2Y purinergic receptor families. P2X receptor activation facilitates synaptic transmission by increasing presynaptic glutamate release probability [92,93], whereas P2Y receptor activation enhances inhibitory effects by restricting presynaptic glutamate release [93,94].

Reactive astrocytes release chemokines and cytokines [95]. For example, astrocytes produce CCL in response to the stimulation of IL-1β and TNF-α [96]. Intrathecal CCL2 causes microgliosis in the spinal cord [97,98]. Mice lacking CCR2- receptors of CCL2 do not show a nerve injury-induced spinal microglial reaction [98]. These effects are likely due to the effect of CCL2 on synaptic transmission. These data strongly suggest that astrogliosis might modulate synaptic efficacy via CCL [99].

Although both astrocytes and microglia are often activated in chronic pain models, there are probably important differences in their pathogenic contribution. For example, astrogliosis but not microgliosis was observed in the SDH of HIV-infected patients [85], indicating their differential contribution to the pathogenesis of HIV-associated pain. In support of this notion, in mouse models of HIV-associated pain, ablation of microglia only affected the early phase of pain development, while suppression of astrogliosis affected both the early and maintenance phases [51]. Moreover, astrocyte inhibitors show efficacy in reducing both early and late phases of neuropathic pain, whereas microglial inhibitors are effective primarily in the early phase [100,101]. These findings collectively show the different temporal roles of microglia and astrocytes in pain pathogenesis. Indeed, microglial activation generally precedes astrocyte activation. Following peripheral nerve injury, spinal microglia rapidly proliferate and reach their highest levels within the first week [102], whereas astrogliosis is observed one week after nerve injury and persists for many months [103,104,105]. Emerging evidence also indicates that microglia may contribute differently to the pathogenesis of different types of pain. Inhibition of microglial activation by minocycline reduces peripheral nerve injury (PNI)-induced neuropathic pain [106]. On the other hand, ablation of microglia by PLX5622 does not affect opioid-induced hyperalgesia [107]. Interestingly, microglia are required for mechanical pain hypersensitivity in male but not female mice [108]. 

## 4. Neural Circuitry Plasticity in the SDH during Pain Pathogenesis

### 4.1. Central Sensitization

Although the enlargement of the receptive field of dorsal horn neurons after peripheral nerve injury has long been observed, it is generally thought to be the enhancement of input from silent or ineffective synapses in periphery but not central neuronal plasticity. Clifford J. Woolf first demonstrated a central component of pain hypersensitivity [109]. Instead of recording dorsal horn neurons, he measured the activity of single biceps femoris α-motoneuron axons as the output of the nociceptive signal since the firing of these neurons in response to nociceptive stimulus explicitly leads to the flex or reflex withdrawal response, and the threshold of the withdrawal response is identical to that of activating pain. Thus, the withdrawal responses are used as a surrogate for pain. This study reveals that, under basal conditions, biceps femoris α-motoneurons display a high firing threshold, and their receptive field is restricted to certain regions such as the toes or hind paws. However, repeated peripheral noxious heat stimuli lead to a reduction in threshold and enlargement of the cutaneous receptive fields from the same neurons, and this effect cannot be reversed by local anesthetic block of the peripheral injury site. These data strongly indicate that noxious heat stimulation induced a central plasticity of the nociceptive system (now termed central sensitization), making it respond to stimuli outside of the injury area and to low-threshold afferents that previously carried innocuous stimuli.

Nowadays, central sensitization is defined as a nociceptive neuronal response in the CNS to normal stimulus or excessive activity in response to painful insults. It is usually caused by increased neuronal excitability, synaptic efficacy, and reduced inhibition. The overall effects of central sensitization include hypersensitivity to stimuli (hyperalgesia), responsiveness to non-noxious stimuli (allodynia), and increased pain response evoked by stimuli outside the area of injury. 

In the spinal cord, SDH neurons subject to central sensitization show one or more of the following alterations: increases in spontaneous activity, reduction in the threshold for activation, increased responses to suprathreshold stimulation, and enlargement of their receptive fields [23]. SDH neurons generally receive effective inputs from small, unmyelinated nociceptors as well as ineffective or subthreshold inputs from large, myelinated fibers. In response to peripheral inflammation, large myelinated fibers, such as Aβ-fibers, begin to release substance P and BDNF, which substantially increases the activities of Aβ-fibers as well as the Aβ-mediated synaptic input to superficial dorsal horn neurons [110,111,112]. Substance P can cause a long-lasting membrane depolarization when binding to NK1 receptors, whereas BDNF can switch inhibitory GABAA receptor-mediated inputs to excitatory [72,113]. Continuous stimulation of these fibers will drive central sensitization. Consequently, the previous subthreshold stimulation from mechanic receptors is able to generate the action potential of superficial dorsal horn neurons, therefore leading to allodynia [114]. In addition, peripheral nerve injury might lead to a degeneration of C fiber terminals in lamina II together with regeneration of injured neurons, including Aβ fibers, providing an opportunity for Aβ fibers to sprout from deep lamina into laminae I-II and impinge on nociceptive-specific neurons. This process will lead to the pain response by the stimuli outside the area of injury [115,116,117,118].

Central sensitization is pivotal in the study of chronic pain pathogenesis. However, the ambiguity needs to be clarified. Central sensitization is a phenomenon that can be explained by a series of cellular or molecular mechanisms. In addition, it is not a universal phenomenon for all the neurons in the nociceptive system. For example, it cannot explain the behavior of inhibitory neurons. The activities of inhibitory neurons, as well as inhibitory synapses, are shown to be reduced during chronic pain. Thus, a more general mechanism underlying chronic pain should be investigated.

### 4.2. Short-Term Synaptic Plasticity 

Short-term plasticity was concurrently recognized by a Chinese and a German group about 80 years ago [119,120]. It was first reported to modulate electrical transmission in the neuromuscular junction, but virtually all types of synapses are regulated by a variety of short-lived and long-lasting processes. Short-term plasticity refers to the change in synaptic strength if a synapse is activated repeatedly within a time scale of several milliseconds to a few seconds. Synaptic strength increases during short-term facilitation and decreases during short-term depression [121]. Short-term plasticity is regulated by both pre- and postsynaptic mechanisms [122,123,124,125,126].

In the nociceptive system, repetitive heat or mechanical stimuli in a few hundred milliseconds resulted in nociceptor sensitization [127,128,129]. Moreover, short-term facilitation is also observed in the synapses from the midline and intralaminar thalamic nuclei (MITN) to the anterior cingulate cortex (ACC) [130]. In the spinal cord, Wind Up is one of the hallmarks of neuronal plasticity during persistent pain. It is characterized by an increased excitability of SDH neurons in response to a low-frequency stimulation train of C fibers. Short-term facilitation might lead to the temporal summation of synaptic potentials, which contributes to the formation of Wind Up [131]. However, since short-term plasticity is recovered in a few seconds, this might not account for the maintenance of chronic pain.

### 4.3. Long-Term Synaptic Plasticity

Different from short-term plasticity, long-term plasticity reflects the synaptic efficacy change in the range from hours to days. It leads to an enhancement in synaptic efficacy (long-term potentiation, LTP) or a decrease in synaptic efficacy (long-term depression, LTD). 

LTP was first observed in the rabbit hippocampus in 1973 [132]. Spinal LTP is reported and regarded as a potential mechanism of pain amplification. For example, high-frequency stimulation (HFS) of the sciatic nerve induced LTP at C-fiber synapses and produced chronic pain-like behavior in mice, with microglia being a particularly important contributor [133]. In addition, low-frequency stimulation (LFS), which is closer to the physiological firing rate of C fibers, is also able to induce LTP on NK1-positive neurons projecting to the periaqueductal grey [134]. In either case, fiber stimulation induces the release of substance P, which binds NK1 receptors on projection neurons in SHD. This process directly potentiates NMDA receptors and leads to a substantial rise in postsynaptic Ca^2+^—an essential process for the development of LTP [135]. 

LTP has been shown to be closely associated with chronic pain. For example, spinal LTP induces hyperalgesia in rats [136]. Using a similar stimulation protocol, the pain perception of human beings is potentiated [137]. On the other hand, the role of LTD in chronic pain is also investigated in rodents and humans. LTD in ACC was substantially impaired in the rat with single-paw digit amputation as well as tail amputation [138,139]. Applying low-frequency stimulation at 1 Hz—the protocol that is used to induce LTD in the spinal cord or cortex—reduces pain perception in humans [137,140]. 

Long-term plasticity can last for hours or up to days and months, which fits the time course of chronic pain. However, long-term plasticity is homosynaptic, meaning that synaptic strength is selectively enhanced in those synapses receiving repetitive stimuli, whereas in central sensitization, both activated synapses and non-activated neighboring synapses are enhanced. In addition, continuous long-term plasticity results in the endless augmentation of synaptic strength, which makes the synapses unstable. 

### 4.4. Homeostatic Plasticity/Synaptic Scaling

LTP and LTD are the two forms of Hebbian plasticity, which refers to a form of activity-dependent synaptic plasticity where concurrent activation of pre- and postsynaptic neurons leads to the strengthening (or weakening) of the connection between the two neurons. However, as mentioned above, continuous involvement of Hebbian plasticity leads to an indefinite change in synaptic strength, resulting in instability. To keep neuronal activity at healthy levels, plasticity must incorporate certain homeostatic control mechanisms. Homeostatic plasticity acts globally in a negative feedback manner to counter the change in synaptic strength [141]. Homeostatic plasticity is not synaptic-specific, and all the synapses on the affected neurons will be regulated. 

Homeostatic plasticity is an important mechanism underlying chronic pain. It has been shown that sustained depolarization of nociceptors strongly reduces the intrinsic excitability of mouse and human DRG neurons [142]. Moreover, spinal cord injury results in an initial activity loss and subsequent hyperexcitability of cortical neurons, consistent with homeostatic activity regulation [143,144,145]. Enhancement of cortical activity diminished injury-induced behavioral hypersensitivity in mice with neuropathic pain [146]. In addition, cortical stimulation techniques, which might restore the initial loss of neuronal activity, have been used for patients with refractory neuropathic pain [147,148]. In the spinal cord, it is recently reported that nerve injury triggers the homeostatic reduction in inhibitory inputs to excitatory interneurons, causing mechanical hypersensitivity and neuropathic pain [149].

### 4.5. Excitation/Inhibition Balance

Gate control theory (Figure 4), formulated in 1965 by Melzack and Wall [150], proposed a role of the ‘gate’ of the spinal cord that controls pain signals to the brain. According to this theory, nociceptors and mechanoreceptors project to the same interneurons and projection neurons in SDH. Both inputs excite projection neurons, but they have opposite effects on inhibitory interneuron activity. Painful signals reduce the activity of inhibitory interneurons, whereas non-painful signals increase their activity. These inhibitory interneurons themselves reduce synaptic transmission between primary afferent and projection neurons. Under normal conditions, mechanoreceptors produce stronger and faster signals than nociceptive stimuli since they are low threshold and their axons are myelinated. In this case, the inhibitory interneurons driven by mechanical signals will block the painful signal, leading to a closed ‘gate’. Loss or reduction of synaptic inhibition will result in an open ‘gate’, allowing pain sensation (Figure 4). 

The spinal cord receives inhibitory inputs, mainly from inhibitory interneurons and the supraspinal regions [151]. Disruption of synaptic inhibition is a common feature of chronic pain resulting from inflammation [152] and nerve injury [42,72,153,154]. For example, GABAergic synaptic transmission is diminished in the spinal cord of neuropathic rats [42]. In addition, peripheral nerve injury results in apoptosis of inhibitory interneurons, which substantially reduce GABAergic and glycinergic inhibitory currents, leading to a state of disinhibition [153]. Intrathecal administration of the GABAA receptor antagonist bicuculline or the glycine receptor antagonist strychnine induces pain behavior [155,156], which is partially due to the facilitated low-threshold input in response to a reduced inhibition. Accordingly, the application of benzodiazepines—agonists of GABAA receptors—exerts clear analgesic actions in animal models of hyperalgesia, as well as in human patients [153,157,158]. Interestingly, although both excitatory and inhibitory neurons receive inhibitory inputs, excitatory neurons rely more heavily on inhibition. They are, therefore, more affected by disinhibition [159].

## 5. Neural Circuit Polarization (NCP): A Novel Form of Neural Circuitry Plasticity in Pain Pathogenesis

It is well known that the overall excitability of neuronal networks in SDH is enhanced in pathologic pain, but the detailed circuitry alteration remains incompletely understood. Recently, it has been demonstrated that pathological pain is associated with not only an increased excitatory input onto excitatory neurons but also a decreased excitatory input onto inhibitory neurons, a process named NCP [51,52]. These polarizations are expected to drive hyperactivation of the nociceptive system and thus result in pathological pain (Figure 5).

NCP was initially observed in HIV-associated pain models. Intrathecal application of gp120—a neurotoxic protein that is specifically associated with HIV-induced pain—results in an increased sEPSC frequency and eEPSC amplitude in excitatory neurons but a decreased sEPSC frequency and eEPSC amplitude in inhibitory neurons. The enhanced excitatory synaptic transmission on excitatory cells and weakened excitatory synaptic transmission on inhibitory cells potentially lead to sensitization of pain circuitry output. In addition, NCP was later observed in opioid-induced hyperalgesia. The application of morphine induces a similar alteration of excitatory inputs on excitatory and inhibitory neurons. Furthermore, a similar phenomenon was observed in chronic constriction injury (CCI)-induced pain [160]. In addition to the spinal cord, NCP might play a role in the pathological condition of the brain. For example, epilepsy can be induced by Kindling, which is, in essence, a process of neuronal sensitization. It is likely that NCP plays a role in the development of epilepsy. Indeed, it has been reported that EPSC to interneurons is weakened in the cortex of patients with temporal lobe epilepsy [161]. Meanwhile, in the animal model of the same disease, EPSCs (mostly to principal cells) in the entorhinal cortex are facilitated [162]. 

In the central nervous system, astrocytes are closely associated with synapses. They directly affect the function of excitatory synapses in several ways, including taking up glutamate, inducing AMPA receptor localization to the postsynaptic density, and releasing gliotransmitters [163,164]. Astrogliosis in SDH has been observed in pathological pain [77,85]. In the model of HIV-associated pain, ablation of astrogliosis via the thymidine kinase/ganciclovir approach sufficiently alleviates gp120-induced hyperalgesia. Meanwhile, this treatment abolishes NCP in the SDH. In addition to HIV-induced chronic pain, astrogliosis is also observed in opioid-induced hyperalgesia (OIH), and it is shown to be important in the development of NCP in this pain model. Namely, the selective block of astrogliosis effectively blocks OIH and NCP [52]. These data suggest that the NCP observed in chronic pain is dependent on astrogliosis, and it is presumably a general mechanism underlying wide-ranging pain models. Previous studies from several labs have shown that astrocytes indeed have dynamic effects on synaptic transmission. On the one hand, astrocytes enhance synaptic transmission by promoting synaptogenesis [164]. On the other hand, it reduces excitatory synaptic activity by facilitating extracellular glutamate and potassium removal during synaptic activity [165]. Furthermore, astrocytes release both glutamate and ATP, which evoke a biphasic regulation of neurotransmitter release in hippocampal synapses [166]. Specifically, astrocytic glutamate has been shown to transiently enhance the probability of neurotransmitter release at CA3-CA1 synapses [167,168,169,170], whereas astrocytic ATP tonically depresses neurotransmission [171]. Interestingly, astrocytes can also biophysically regulate inhibitory transmission [172,173,174]. 

Although astrocytes are the main contributor to NCP and hyperalgesia, the role of microglia in NCP cannot be excluded. Ablation of microglia partially inhibits the early phase of gp120-induced hyperalgesia, indicating a role of microglia in the early expression of NCP. A similar effect of microglial-derived BDNF in the induction of NCP-like phenomena is reported in rat SDH [160].

Conditional knockout of *Wnt5a* in neurons or *ROR2* in astrocytes blocks astrogliosis as well as pathogenic pain, indicating an important role of neuron-to-astrocyte Wnt5a-ROR2 signaling in the expression of NCP. We believe that Wnt5a signaling induces NCP via activating astrocytes. However, we cannot exclude the possibility that Wnt5a directly affects synaptic transmission. Wnt5a has been shown to enhance both excitatory and inhibitory synaptic transmission in hippocampal neurons [175,176], but how Wnt5a affects excitatory transmission in inhibitory neurons remains unclear. 

IL-1β is crucial in the expression of NCP. Application of gp120 or morphine results in an increase in IL-1β expression in SDH. Ablation of astrogliosis blocks this upregulation. Moreover, IL-1β receptor antagonist IL-1Ra blocks gp120- or morphine-induced pain and NCP. These data indicate the critical role of astrocytic IL-1β in NCP. As a potent inflammatory cytokine, IL-1β has been generally shown to induce synapse loss and impair synaptic plasticity [177,178]. However, it has been reported that IL-1β does show a dual modulation on excitatory synapses. In the nucleus tractus solitarius (NTS) of the brain stem—a key nucleus for immune-to-brain signaling—IL-1β produces PGE2, which increases the sEPSC frequency of local NTS neurons and decreases the amplitude of eEPSCs from the vagal afferent terminal simultaneously (Vincent Marty, EJN, 2008). Further investigation is needed to understand whether IL-1β also produces PGE2 in SDH and, in turn, has the opposite effect on excitatory synapses in excitatory and inhibitory cells. In addition, it has been shown that BNDF also induces NCP in SDH [160], and BDNF has the opposite effect on excitatory synapses in excitatory and inhibitory cortical neurons, similar to that of NCP [179]. Therefore, it is worthy to investigate whether IL-1β and BDNF share similar mechanisms in the induction of NCP. In addition, since astrogliosis is critical in the expression of NCP and pain pathogenesis, it is, therefore, reasonable to speculate that other signaling pathways associated with astrogliosis might also contribute to the expression of NCP. For example, SCI leads to the release of several proinflammatory cytokines—TNF-α and IL-6 [180,181]—which then activate a number of signal transduction pathways such as STAT3, NF-kB or MAPK [182,183]. It is, therefore, interesting to investigate their roles in the expression of NCP.

These findings about NCP in different pain models indicate that NCP is a general form of pain circuitry plasticity contributing to the development of different types of pathological pain. It will be of interest to examine the expression of NCP in other models, such as CCI and SNI neuropathic pain models.

## 6. Comparison of NCP with Other Types of Circuitry Plasticity in Chronic Pain Models

Chronic pain is strongly associated with central sensitization, which represents a phenomenon that includes increased responsivity and lowered threshold. NCP provides a cellular mechanism for central sensitization. As mentioned above, NCP is manifested as an increased excitatory input in excitatory cells and a decreased excitatory input in inhibitory cells. The expected net effect is enhanced activation of nociceptive projection to the brain since both excitatory and inhibitory interneurons send their outputs to projection neurons. As an umbrella term, central sensitization usually implies the hyperactivation of spinal neurons. NCP is consistent with the idea of central sensitization in that it shows enhanced excitatory inputs to excitatory neurons. However, NCP also reveals hypoactivation of excitatory transmission to inhibitory neurons. Hence, the discovery of NCP indicates that the development of pathological pain, such as HIV-1 gp120-induced pain and OIH, is likely associated with both sensitization of excitatory neurons and desensitization of inhibitory neurons rather than universal sensitization of all spinal neurons. 

Previous studies show LTP is a critical form of synaptic plasticity underlying the expression of pathological pain [134,184,185,186]. The observation of increased EPSCs in excitatory neurons during NCP expression indicates that the pathogenic LTP is expressed at excitatory synapses between excitatory neurons. It is currently unclear whether the NCP-associated increase in EPSCs in excitatory neurons can causally contribute to LTP expression or is simply a result of LTP. In addition, the expression of NCP is also associated with decreased EPSCs in inhibitory neurons. This finding indicates a diminished excitatory synaptic transmission onto inhibitory neurons. Therefore, it is tempting to predict that LTD is expressed at excitatory synapses onto inhibitory neurons in the SDH of pain models. Although the involvement of LTD in pain development has not been reported, the discovery of NCP suggests the development of pathological pain is associated with the expression of both LTP and LTD in SDH neural circuits at excitatory synapses on excitatory neurons and inhibitory neurons, respectively. 

Maintaining balanced excitatory and inhibitory synaptic inputs, commonly known as excitation and inhibition balance (E/I balance), is a critical aspect of the homeostatic state of neural circuits, which is crucial for the physiological function of the circuits [187]. Dysregulated glutamatergic and/or GABAergic neurotransmission can interrupt E/I balance. E/I imbalance, such as a decrease in inhibitory tone in the SDH, has been suggested to play a crucial role in pain pathogenesis [188,189]. For example, in a neuropathic pain model, Cao et al. show a reduction in inhibitory inputs to excitatory interneurons in the spinal cord [149]. However, the circuitry mechanism by which E/I imbalance is expressed during pain pathogenesis is unclear. The discovery of NCP suggests that polarized excitatory inputs to excitatory and inhibitory neurons, namely the increase in excitatory inputs to the former, while the decrease in the excitatory inputs to the latter, provides a probable mechanism of pain-related E/I imbalance.

In summary, NCP probably integrates with different types of plasticity in the pain circuits and facilitates their expression. It is tempting to speculate, for instance, that at the initial stage of NCP expression, it may contribute to the development of short-term facilitation, leading to Wind-up. The continuous expression of NCP may contribute to the development of long-lasting synaptic plasticity in pain circuits, such as LTP at excitatory synapses on excitatory neurons and LTD at excitatory synapses on inhibitory neurons. Meanwhile, NCP expressed at both its early and maintenance stages may cause an E/I imbalance, resulting in central sensitization of the pain circuitry. If these conceived integrations are validated, NCP may provide a general circuitry mechanism to support the expression of various forms of circuitry plasticity underlying pain pathogenesis and contribute to specific aspects of pain circuitry malfunction mediated by these forms of plasticity. 

## 7. Conclusions

Pathological pain is the result of the alteration of multiple neural circuits. The recently discovered NCP is likely a fundamental mechanism of circuitry plasticity underlying pain pathogenesis. NCP is manifested as enhanced excitatory inputs onto excitatory neurons and weakened excitatory inputs onto inhibitory neurons. The polarization of excitatory drive on excitatory and inhibitory neurons during the expression of NCP is expected to lead to an increase in overall excitatory tone and a decrease in inhibitory tone in the SDH pain circuits. Thus, NCP likely contributes to the expression of E/I imbalance, as well as the central sensitization. Further study on the molecular and synaptic processes underlying NCP expression will shed light on novel circuitry mechanisms of pain pathogenesis.

## Figures and Tables

**Figure 1 cells-13-00398-f001:**
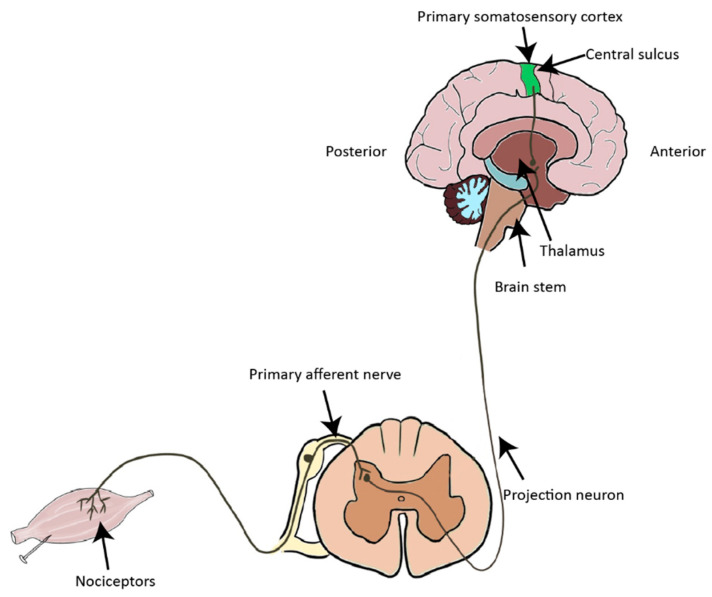
Diagrammatic outline of the ascending pain pathways. Noxious stimuli are transformed into electrical signals in nociceptors and then transmitted to the spinal cord, where the signals are decoded and processed. The nociceptive signals are converged onto projection neurons, which relay the nociceptive information to the brain stem and the thalamus.

**Figure 2 cells-13-00398-f002:**
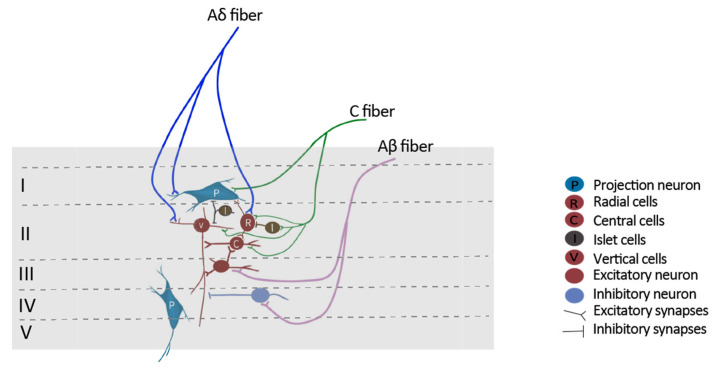
Illustration of simplified pain neuronal circuits in the spinal dorsal horn (SDH). Nociceptors (Aδ and C fibers) convey pain signals from the peripheral to projection neurons, which are mainly in lamina I of the SDH and transmit the signal to the supraspinal levels and the brain. Nociceptive fibers (Aδ and C) synapse with projection neurons in lamina I and interneurons in lamina II. Interneurons interact with and modulate the activity of projection neurons in lamina I via monosynaptic or polysynaptic pathways. Non-nociceptive Aβ fibers project mainly to deep layers in the SDH (laminae III–V). The descending pathways are not shown here.

**Figure 3 cells-13-00398-f003:**
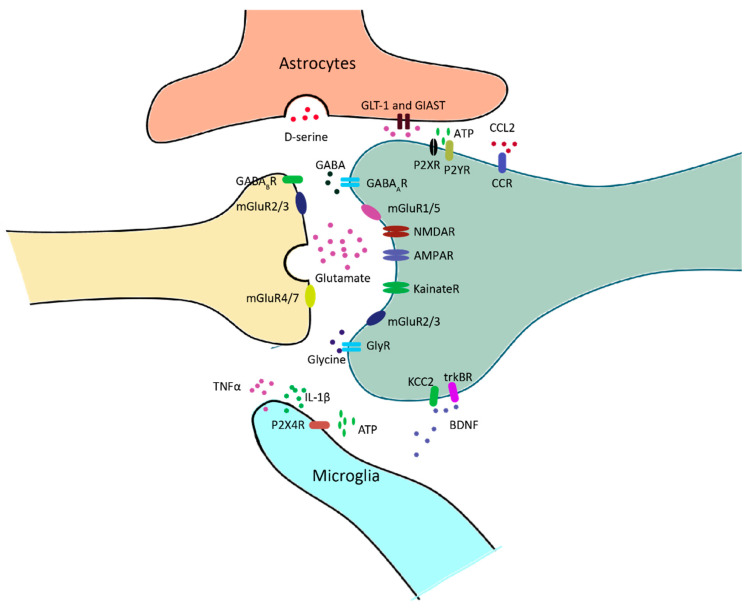
Neuron–glia interactions at nociceptive synapses in the spinal dorsal horn (SDH). Glutamate or GABA/Glycine is released upon the arrival of action potential at the presynaptic fiber. GABA and glycine receptors mediate inhibitory transmission, and glutamatergic receptors mediate excitatory transmission. Ionotropic glutamate receptors, including AMPA receptors, NMDA receptors and kainate receptors, are mainly located in the postsynaptic domain. Metabotropic glutamate receptors are located in both the pre- and postsynaptic domains. Excessive glutamate is taken up by astrocytes and converted to glutamine for the synthesis of glutamate. In response to neuropathy, microglia release a series of inflammatory factors, such as TNFα and IL-1β, which modulate the activity of neurons via a neuro-immune interaction. BDNF is also released by microglia, and it affects neuronal excitability by regulating Cl- homeostasis.

**Figure 4 cells-13-00398-f004:**
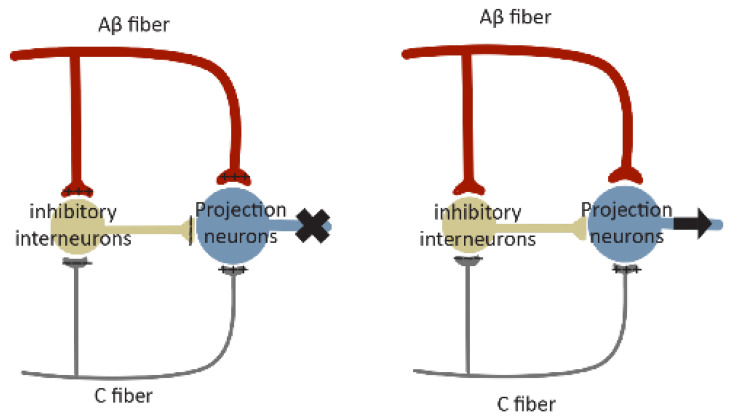
Schematic diagram of gate control theory. Projection neurons receive excitatory inputs (indicated by ‘+’ signs) from large Aβ fibers and small C fibers, and inhibitory inputs (indicated by ‘−‘ signs) from inhibitory interneurons. Interneurons are excited by Aβ fibers but inhibited by C fibers. The balance of large Aβ fibers and small C fibers determines the output of inhibitory interneurons. When inputs from Aβ fibers are stronger, the inhibitory interneurons are excited, resulting in a closed ‘gate’ to inhibit projection neuron activation (indicated by ‘X’ in the **left panel**). When the inhibitory inputs from C fibers are stronger, the activity of inhibitory interneurons will be reduced, leading to an open ‘gate’ to permit projection neuron activation (indicated by the arrow in the **right panel**).

**Figure 5 cells-13-00398-f005:**
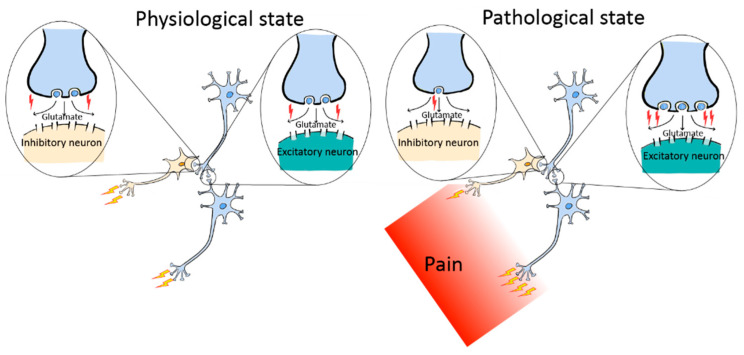
Illustrations of NCP expression at excitatory synapses on excitatory and inhibitory neurons in neuronal circuits of the SDH at physiological (**left**) and pathological (**right**) states. At physiological conditions, the circuits are in homeostatic states, and the activity of these types of excitatory synapses (indicated by red flashes) is stable. In pathological states (**right panel**), when the circuitry homeostasis is disrupted and NCP is expressed, excitatory inputs to inhibitory neurons are weakened, whereas excitatory inputs to excitatory neurons are strengthened. Consequently, the outputs (indicated by yellow flashes) of the inhibitory neurons are expected to decrease while the outputs of excitatory neurons increase. This form of circuitry plasticity will result in E/I imbalance and the expression of pathological pain.

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
