# Peer review of "Neural Circuitry Polarization in the Spinal Dorsal Horn (SDH): A Novel Form of Dysregulated Circuitry Plasticity during Pain Pathogenesis"

_cells, 2024, doi:10.3390/cells13050398_

Round 1
Reviewer 1 Report
Comments and Suggestions for Authors
The problem of studying the mechanisms of formation of neuropathic pain syndrome is certainly relevant. A large number of scientific groups around the world are engaged in the study of this pathology. The authors presented a well-written review characterizing an additional mechanism for the development of the nociceptive reaction. I recommend this manuscript for publication in the journal Cells.
A few minor notes:
1. In Figure 2 there is no designation of an inhibitory neuron (blue oval).
2. Since the authors chose a broad topic for the review article, the list of references used is quite impressive. However, I would suggest that the authors pay additional attention to studies of neuropathic pain in the chronic constriction injury (CCI) and spared nerve injury (SNI).
Author Response
Reviewer 1
The problem of studying the mechanisms of formation of neuropathic pain syndrome is certainly relevant. A large number of scientific groups around the world are engaged in the study of this pathology. The authors presented a well-written review characterizing an additional mechanism for the development of the nociceptive reaction. I recommend this manuscript for publication in the journal Cells.
A few minor notes:
- In Figure 2 there is no designation of an inhibitory neuron (blue oval).
We thank Reviewer 1 to point out the glitch we made in figure 2. Indeed, the inhibitory neuron is mislabeled. We added the missed designation of an inhibitory neuron in Figure 2.
- Since the authors chose a broad topic for the review article, the list of references used is quite impressive. However, I would suggest that the authors pay additional attention to studies of neuropathic pain in the chronic constriction injury (CCI) and spared nerve injury (SNI).
We thank the reviewer’s comment on the focus of the manuscript. Additional relevant work was considered on CCI and SNI (line 368). In addition, because the current knowledge of NCP has mainly obtained from HIV-associated pain and OIH, we acknowledge a potential interesting line of research of investigating NCP expression in CCI and SNI models (line 432).
Reviewer 2 Report
Comments and Suggestions for Authors
The review of Chang and Tang suggest that pain generation may appears, at least partially by a patho-genesis-neural circuitry polarization (NCP). It consists in an unbalance of excitatory/inhibitory inputs to glutamatergic and GABAergic neurons in the spinal cord. The participation of the glia is also crucial to this polarization. This review is interesting, but you should pay attention to some of these comments:
There is an imbalance between the first part, which deals with a review of the synaptic mechanisms in the spinal cord, well-known results, and its novel proposal about “NCP: a novel form of neural circuitry plasticity implicated in pain pathogenesis”. I suggest reducing the first part of the review to indicate the fundamental aspects of pain transmission in the spinal cord and giving more importance to your proposal.
- Please pay attention to some grammatical errors. The text should be reviewed by an expert. For example, in the Abstract “Virous forms of circuitry plasticity,”. I assume you are referring to Various. There are also many places where the full stop does not have a space with the previous word or parenthesis; for example, impermeable).Almost all the
- You refer in the text and Figure 1 that the pain pathway reaches the thalamus and forgets about the cortex. Please indicate in the text the cortical areas related to pain sensation and include them in the figure.
-Another inaccuracy is the following sentence: “In the central nervous system (CNS), spinal dorsal horn is the first relay center of nociceptive information”. You forget that the trigeminal nucleus receives all the somesthetic information from the face, including pain.
- The font size of the word Glutamate is very small in Figure 5.
-Lines 469-471, it is not correct to include unpublished data in a review. Please, remove it.
- 9. Conclusions. Taking into account that NCP is the newest part of the manuscript, I would include a more detailed description of what it consists of.
Comments on the Quality of English LanguagePlease pay attention to some grammatical errors. The text should be reviewed by an expert. For example, in the Abstract “Virous forms of circuitry plasticity,”. I assume you are referring to Various. There are also many places where the full stop does not have a space with the previous word or parenthesis; for example, impermeable).Almost all the
Author Response
Reviewer 2
The review of Chang and Tang suggest that pain generation may appears, at least partially by a patho-genesis-neural circuitry polarization (NCP). It consists in an unbalance of excitatory/inhibitory inputs to glutamatergic and GABAergic neurons in the spinal cord. The participation of the glia is also crucial to this polarization. This review is interesting, but you should pay attention to some of these comments:
There is an imbalance between the first part, which deals with a review of the synaptic mechanisms in the spinal cord, well-known results, and its novel proposal about “NCP: a novel form of neural circuitry plasticity implicated in pain pathogenesis”. I suggest reducing the first part of the review to indicate the fundamental aspects of pain transmission in the spinal cord and giving more importance to your proposal.
We thank Reviewer 2 for the criticism on the structure if the manuscript. We agree that the introduction of general synaptic mechanism of chronic pain is overly weighted. We reduced this part by refining the excessive introduction of less related knowledge. For example, we shorted the description of anatomy of nociceptive pathway. We also summarized the most information of excitatory and inhibitory receptors instead of elaborate the structure of each receptor.
Please pay attention to some grammatical errors. The text should be reviewed by an expert. For example, in the Abstract “Virous forms of circuitry plasticity,”. I assume you are referring to Various. There are also many places where the full stop does not have a space with the previous word or parenthesis; for example, impermeable).Almost all the
We also thank Reviewer 2 to point out the grammatic errors we made. We carefully inspected the manuscript and corrected all the grammatic errors.
- You refer in the text and Figure 1 that the pain pathway reaches the thalamus and forgets about the cortex. Please indicate in the text the cortical areas related to pain sensation and include them in the figure.
The cortical destination of pain signal is missing. We added that information in Figure 1.
-Another inaccuracy is the following sentence: “In the central nervous system (CNS), spinal dorsal horn is the first relay center of nociceptive information”. You forget that the trigeminal nucleus receives all the somesthetic information from the face, including pain.
We added that information in the text (Line 39).
- The font size of the word Glutamate is very small in Figure 5.
The font size in Figure 5is changed as suggested.
-Lines 469-471, it is not correct to include unpublished data in a review. Please, remove it.
We removed the description of unpublished data.
- 9. Conclusions. Taking into account that NCP is the newest part of the manuscript, I would include a more detailed description of what it consists of.
We included a more detailed description of NCP in the conclusion.
Please pay attention to some grammatical errors. The text should be reviewed by an expert. For example, in the Abstract “Virous forms of circuitry plasticity,”. I assume you are referring to Various. There are also many places where the full stop does not have a space with the previous word or parenthesis; for example, impermeable).Almost all the
We carefully inspected the manuscript and corrected all the grammatic errors.
Round 2
Reviewer 2 Report
Comments and Suggestions for Authors
The authors have changed the text according to my suggestions. I have no further comment on the text. However, Figure 1 is unacceptable for publication.
In the figure there is an axon that reaches the thalamus but from here a neuron whose axon ends in the S1 cortex would depart. The cortex is the entire top of the drawing and not just the small area (green) indicated by an arrow.
If the vertical line in the cortex indicates the central sulcus, then the S1 cortex (primary somatosensory cortex) is more caudal to that sulcus and not rostral as indicated by the authors. Also, the arrow indicating the thalamus should indicate the central part of the thalamus and not just a more anterior part that is unclear whether it is thalamus or hypothalamus.
Therefore, this drawing has very serious errors and cannot be published without being corrected.
Author Response
Dear Editor,
We thank Reviewer 2 again for his/her valuable comments. As he/she correctly pointed out, the green area in figure 1 should be labelled more precisely, and it should be more caudal to the central sulcus. We now moved it to the correct area and named it more precisely as “primary somatosensory cortex”. In addition, we marked the anterior and posterior direction of the brain to make the navigation clearer. Furthermore, we drew a neuron in the thalamus sending its axon to S1. Last but not least, we moved the arrow indicating thalamus to the central part, as it easily leads to misunderstanding in the previous position.